

# Time to cash in on positive interactions for coral restoration

Elizabeth C. Shaver and Brian R. Silliman

Division of Marine Science and Conservation, Nicholas School of the Environment, Duke University, Beaufort, NC, United States of America

## ABSTRACT

Coral reefs are among the most biodiverse and productive ecosystems on Earth, and provide critical ecosystem services such as protein provisioning, coastal protection, and tourism revenue. Despite these benefits, coral reefs have been declining precipitously across the globe due to human impacts and climate change. Recent efforts to combat these declines are increasingly turning to restoration to help reseed corals and speed-up recovery processes. Coastal restoration theory and practice has historically favored transplanting designs that reduce potentially harmful negative species interactions, such as competition between transplants. However, recent research in salt marsh ecosystems has shown that shifting this theory to strategically incorporate positive interactions significantly enhances restoration yield with little additional cost or investment. Although some coral restoration efforts plant corals in protected areas in order to benefit from the facilitative effects of herbivores that reduce competitive macroalgae, little systematic effort has been made in coral restoration to identify the entire suite of positive interactions that could promote population enhancement efforts. Here, we highlight key positive species interactions that managers and restoration practitioners should utilize to facilitate the restoration of corals, including (i) trophic facilitation, (ii) mutualisms, (iii) long-distance facilitation, (iv) positive density-dependence, (v) positive legacy effects, and (vi) synergisms between biodiversity and ecosystem function. As live coral cover continues to decline and resources are limited to restore coral populations, innovative solutions that increase efficiency of restoration efforts will be critical to conserving and maintaining healthy coral reef ecosystems and the human communities that rely on them.

## INTRODUCTION

Coral reefs are one of the most biodiverse and productive ecosystems on Earth, and provide critical services to at least 500 million people throughout the world (*Wilkinson, 2004*). In addition to supporting healthy and thriving human communities through food provisioning, coastal protection, and tourism revenue, coral reefs are essential ecosystems within the landscape of coastlines, facilitating seagrass, mangrove, and terrestrial habitats (*Dorenbosch et al., 2005*; *Mumby et al., 2004*). Despite the many benefits to both natural and human communities, coral reefs around the world are rapidly degrading due to a combination of human activities and climate change. In response to these losses, conservation

Corresponding author
Elizabeth C. Shaver, ecs39@duke.edu

efforts are increasingly turning to restoration as a strategy to promote coral reef recovery by transplanting colonies that will hopefully grow, reproduce, and reseed reefs (*Johnson et al., 2011*; *Young, Schopmeyer & Lirman, 2012*). Although active coral transplantation efforts are increasing across the globe (*Lirman & Schopmeyer, 2016*), restoration remains costly (median, $162,455 US per ha) and most projects are done on scales temporal (1 yr) and spatial scales (<1 ha) (*Bayraktarov et al., 2016*). Transplant survivorship, however, has been high relative to other habitats (64.5% survival) (*Bayraktarov et al., 2016*); thus, coral restoration shows promise but requires further innovative approaches that improve restoration efficiency. Increased yield for each conservation dollar spent will be critical if we are to reach restoration scales needed to rehabilitate large enough coral populations that restore coral reef processes and services at the ecosystem level.

For many decades, the paradigm in coastal restoration has been to minimize negative interactions (e.g., competition) between transplant neighbors (*Silliman et al., 2015*). This paradigm was transferred directly from forestry science, which developed in the late 1940s (*Halpern et al., 2007*). However, positive interactions are common in marine systems and often play pivotal roles in ecosystem development and recovery (*Bruno & Bertness, 2001*). For instance, a mutualism between salt marsh grasses and ribbed mussels is key to salt marsh resilience and recovery after severe drought events, as mussel mounds increase nutrients and water retention that reduces marsh grass mortality during drought (*Angelini et al., 2016*). In kelp forests, predator populations promote ecosystem resistance and recovery by indirectly facilitating kelp populations through a trophic cascade (*Estes & Palmisano, 1974*). Positive species interactions in coral reef ecosystems could likewise be identified and used to assist in the active recovery of coral populations and habitats.

With increased consideration of facilitation in ecological theory (*Bruno, Stachowicz & Bertness, 2003*), recent papers have also recently made the case for systematically including positive interactions in aquatic and coastal restoration (*Halpern et al., 2007*; *Gedan & Silliman, 2009*). Unfortunately, a decade later, a survey of coastal wetland conservation agencies found that reducing negative interactions was still the predominant focus in wetland restoration designs (*Silliman et al., 2015*). Subsequent experimental studies showed that altering restoration practices to instead harness positive species interactions increased transplant survivorship by ∼100% and biomass by ∼200% with no additional cost (*Silliman et al., 2015*). Efforts in coral reef restoration, however, have not been made to systematically use facilitation in restoration designs, even though numerous positive interactions have been identified in ecological studies (e.g., herbivores that facilitate corals by suppressing algae, crustose coralline algae that promotes coral settlement). Indeed, a Web of Science search for coral restoration studies from 1980–2016 (generating 104 studies; see Fig. 1 for search terms) shows very few studies (10 studies, 9.6%) have examined positive interactions in a restoration context, with most of these papers focusing on herbivore-algae interactions (6 studies, Fig. 1). Below, we highlight positive interactions that naturally occur on coral reefs around the world that may be available to use to enhance coral restoration success and thus increase the efficiency and scale of these efforts.

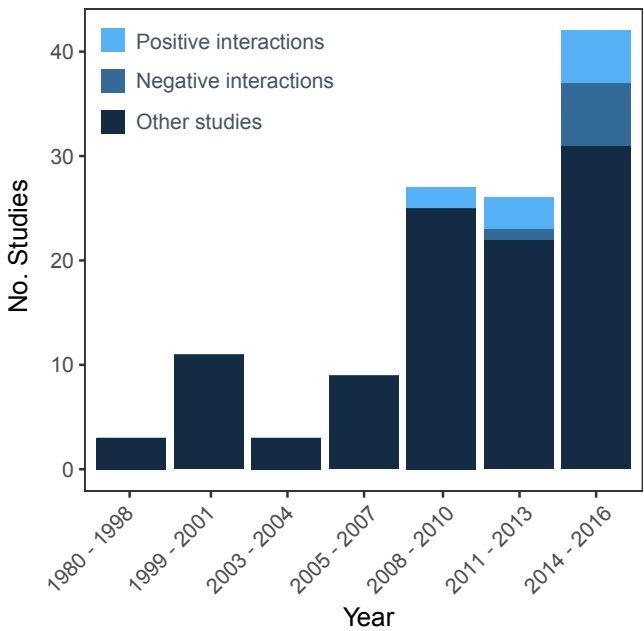

**Figure 1** Number of coral restoration studies from a Web of Science search for TOPIC: ("Coral Restoration"), OR TOPIC: ("Coral Propagation") OR TOPIC: ("Coral Gardening") OR TOPIC: ("Coral Nurseries") from 1980–2016 that examine negative interactions (i.e., competition, predation) and positive interactions (i.e., facilitation, mutualism, cooperation) or neither.

## TROPHIC FACILITATION

Trophic facilitation occurs when one species is positively impacted through the feeding activities of another species. One example includes trophic cascades where predators, by suppressing densities of primary consumers, can increase densities of basal prey species such as plants (e.g., effects of wolves on shrub communities in the Great Lakes: *Callan et al., 2013*; effects of otters on kelp forests in the Pacific: *Estes & Palmisano, 1974*). Another common example occurs when consumers facilitate species that are competitively inferior. For instance, ungulates in North America (*Beschta & Ripple, 2009*) and elephants in Africa (*Sinclair et al., 2010*) can facilitate grassland persistence by preferentially feeding on saplings, thereby suppressing the growth of competitively dominant trees.

In coral reefs, the facilitation of corals by herbivores (via suppressing the overgrowth of macroalgae) has been studied for decades. Results of comparative and experimental studies have shown that herbivorous fish and urchins are critical for the success of corals, and this positive interaction is general across almost all regions where corals occur (*Ogden & Lobel, 1978*; *Hay, 1984*; *Mumby et al., 2006*; *Burkepile & Hay, 2008*). Because of this research, coral conservation programs often focus on harnessing the positive impacts of herbivores on corals (e.g., NOAA *Acropora* Recovery Plan: *National Marine Fisheries Service, 2015*), and many restoration manuals likewise recommend transplanting corals in areas with high herbivore densities, such as in marine protected areas (MPAs; Caribbean *Acropora* Restoration Guide: *Johnson et al., 2011*) (Fig. 2). Despite the incorporation of these ecological findings into restoration, no studies to date have experimentally evaluated

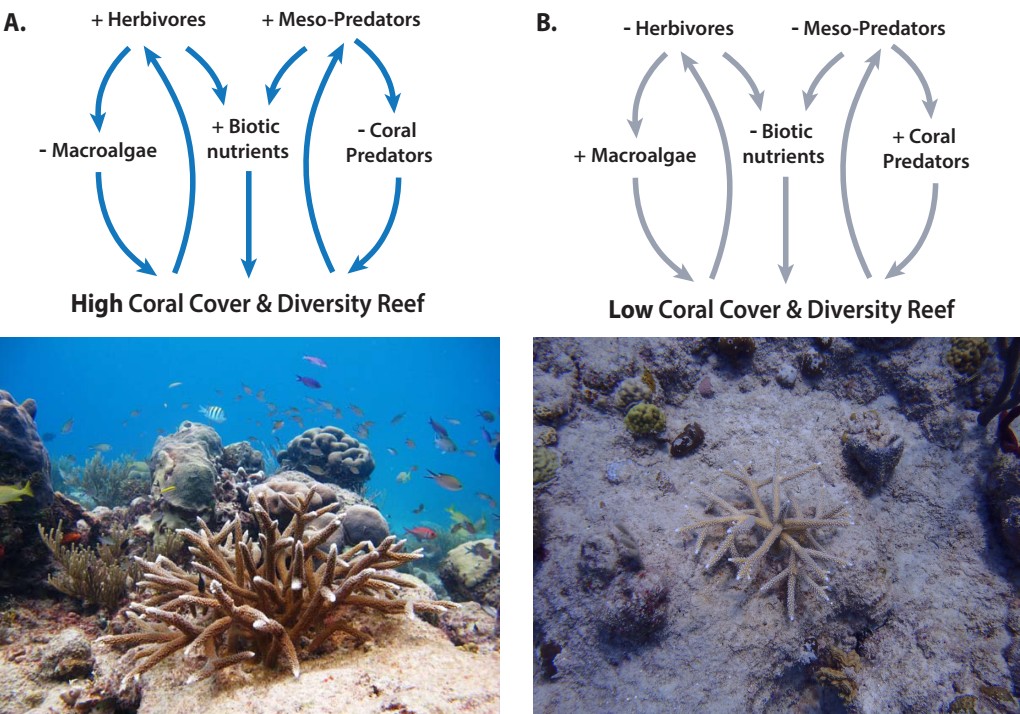

**Figure 2** **Trophic interactions and feedback loops between corals and reef inhabitants.** (A) Reefs with high coral cover and diversity support higher levels of herbivores (e.g., fish and urchins) that reduce macroalgae and meso-predators. Meso-predators reduce populations of corallivores, and herbivores reduce macroalgae, which both promote increased coral cover. Additionally, herbivores and meso-predators produce more biotic-derived nutrients that enhance coral growth. These effects all facilitate healthy coral communities. (B) Reefs with low coral cover and diversity support fewer herbivores and meso-predators, leading to increased macroalgae and corallivores but less biotic nutrients. These effects further reduce coral cover. Photo credits: (left) Kemit Amon-Lewis, The Nature Conservancy; (right) Elizabeth Shaver, Duke University.

the effects of coral restoration success in areas with high vs. low herbivore abundance. This work is ultimately important as many reef herbivores can negatively affect corals or reef systems by acting as corallivores or bioeroders of reef framework. Recent research suggests that grazing may be more detrimental on degraded reefs with few living corals and reduced reef accretion rates (*Rotjan & Lewis, 2008*; *Mumby, 2009*; *Kuffner & Toth, 2016*). Therefore, taking advantage of ecological interactions that facilitate corals for restoration must also include rigorous evaluation of their effects and can include assessing threshold effects or evaluating how different species vary in their functional roles (as suggested by *Rotjan & Lewis, 2008*). In addition to macroalgae, future restoration designs may also benefit from extending this research to examine how trophic interactions also influence other benthic coral competitors such as sponges, fire corals, and mat-forming zoanthids.

In contrast to the indirect effects of herbivores on corals, trophic cascades and direct predation on corals have been considered within the context of restoration and conservation to a far lesser extent. There is experimental and observational evidence, however, to suggest that trophic cascades and food web linkages can positively affect coral populations. For

instance, experimental removals of pufferfish in the Caribbean showed that these fish can facilitate soft corals by reducing populations of corallivorous flamingo tongue snails (*Burkepile & Hay, 2007*). Similarly, the breakdown of trophic cascades due to overfishing has also been suggested to lead to population outbreaks of corallivores such as nudibranchs in Hawaii (*Gochfeld and Aeby 1997*), *Drupella* snails in the Western Indian Ocean (*McClanahan, 1997*), and crown-of-thorns starfish in Fiji (*Dulvy, Freckleton & Polunin, 2004*). No research has yet been attempted to evaluate the positive effects of trophic cascades on coral restoration success, even though the potential to enhance restoration efficiency is high given that corallivores are common sources of mortality for new coral transplants (*Schopmeyer & Lirman, 2015*; *Lirman & Schopmeyer, 2016*). Additionally, much time and manpower are currently being invested in the Caribbean to manually remove corallivores from transplants (*Williams et al., 2014*; *National Marine Fisheries Service, 2015*). Like herbivores, one of the best places to examine these effects may be within effectively managed MPAs where predators of corallivores should theoretically be higher (Fig. 2).

Beyond tri-trophic facilitation of corals through predator control of corallivores, a more general discussion has emerged over whether top predators can create interactions that trickle down to positively affect coral populations. The evidence so far has been equivocal, as recent observational research correlating top predators with coral or algal-dominated systems have found varying results (*Sandin et al., 2008*; *Ruppert et al., 2013*) potentially caused by the fact that top predators often feed on multiple trophic levels (*Valentine & Heck, 2005*). Despite this, more direct trophic linkages may be found that can be used specifically for restoration when corals are most vulnerable (i.e., when they are first transplanted and small), for instance protecting a predator that strongly controls populations of a particular corallivore. The ambiguity and paucity of experiments in this research avenue, and therefore its use in informing coral restoration practices, highlights the great need for more research on how complex food web interactions can facilitate coral populations.

## MUTUALISMS

Mutualistic interactions, or reciprocal positive interactions between species, are fundamental to the success and persistence of foundation species in many marine and terrestrial ecosystems. For example, mutualisms between plants and mycorrhizal fungi underlie forests (*Dighton & Mason, 1985*) and salt marshes (*Daleo et al., 2007*) by facilitating plant colonization in otherwise stressful environments. Other reciprocal positive interactions facilitate the persistence of foundation species by protecting them from natural enemies. An example of this is the symbiosis between ants and Acacia trees, where ants protect trees from antagonistic species (stem boring beetles: *Palmer et al., 2008*) and pathogens (*González Teuber, Kaltenpoth & Boland, 2014*) in exchange for refuge and habitat. Similarly, the formation of coral reefs throughout the tropics would not occur without a mutualism between corals and photosynthetic algae (*Symbiodinium*), which provide substantial nutrition to corals and enhance skeletal deposition (*Goreau & Goreau, 1959*). Because of the high biodiversity on coral reefs, there are likely many reciprocal positive interactions that promote coral success and that can be identified for use in restoration designs.

The relationship between corals and crustose coralline algae (CCA) is one of the most well-known reciprocal positive interactions that can encourage healthy coral reef functioning. While coral reefs provide habitat, CCA is critical for cementing and stabilizing reef structure and facilitating settlement of coral larvae (*Heyward & Negri, 1999*). Reef-dwelling sponges are also important for substrate stabilization as well as food web nutrient cycling through the conversion of detritus into food resources for reef consumers (*De Goeij et al., 2013*). Other mutualisms can increase restoration efficiency by reducing the harmful effects of natural enemies on corals. In French Polynesia, for example, amphipods that live in corals of the genus *Montipora* protect corals from predatory seastars (*Bergsma & Martinez, 2011*; *Bergsma, 2012*). Similarly, mutualistic crabs in the Pacific increase the survivorship and growth of branching corals by reducing sedimentation (*Glynn, 1976*), predation by crown-of-thorns seastars (*Pratchett, Vytopil & Parks, 2000*; *Pratchett, 2001*), and vermetid snail effects (*Stier et al., 2010*). Many fish species that use coral reefs for habitat can also promote coral growth by enhancing the transfer of nutrients in areas where macroalgae are trophically controlled (*Burkepile et al., 2013*).

Despite these many examples of mutualisms that support coral success, few studies to date have examined or experimented with incorporating species-specific mutualisms into restoration. One recent study that seeded a reef with sponges prior to transplanting corals found significant increases in rubble consolidation that in turn enhanced coral survivorship in that environment (*Biggs, 2013*). Similarly, an experiment that explicitly transplanted corals in areas with high and low abundances of grunts showed that fish significantly increased coral survivorship and growth through nutrient transfer (*Shantz et al., 2015*). These studies reveal that reciprocal positive relationships can be used in innovative and strategic ways in a restoration context. Thus, future restoration designs across regions should seek to identify local mutualisms that can be harnessed to increase success naturally without additional manpower.

## LONG-DISTANCE FACILITATION

Positive interactions can also occur between species that are not in contact but separated by distances of tens or thousands of meters (*Van de Koppel et al., 2015*). Long-distance, positive impacts can be generated by the amelioration of physical and/or biological stress. For instance, in coastal habitats the structure of intertidal oyster reefs reduces wave energy and erosion allowing mud flats to accrete behind reefs, which then provide a suitable environment for the development of relatively wave-intolerant salt marsh grasses (*Meyer, Townsend & Thayer, 1997*). Long-distance facilitation can also affect the spatial organization of ecosystems, such as bands of mussel beds that form across large mud flats due to the interaction between local facilitation of mussels but resource competition over larger areas (*Van de Koppel et al., 2015*). As coral reefs are often biologically and physically linked to adjacent tropical habitats, we examine potential long-distance facilitations that may be useful for coral restoration.

Decades of research has shown that the health, productivity, and biodiversity of coral reefs is directly related to their proximity to tropical seagrass meadows and mangrove forests.

For example, both habitats facilitate corals by reducing stressors such as sedimentation or nutrient pollution from coastal development and runoff that may smother corals or enhance algal overgrowth (*Christianen et al., 2013*; *Storlazzi et al., 2011*). Seagrass meadows are also important regulators of water quality, with recent surveys finding 2-fold less coral disease on reefs near seagrasses (*Lamb et al., 2017*). Coral productivity may be enhanced by naturally-derived nutrients from seagrasses or mangroves through detritus or the excrement of animals that forage in those habitats but live on coral reefs (e.g., grunts: *Shantz et al., 2015*) (Fig. 2). In addition to trophic linkages, seagrasses and mangroves can enhance the biomass and diversity of many reef-associated species by acting as a nursery habitat during critical early life stages (*Nagelkerken et al., 2002*) (Fig. 3). Rainbow parrotfish, for example, are important large herbivores in the Caribbean that are significantly higher on coral reefs next to healthy mangrove habitats (*Mumby et al., 2004*), and in general the biomass of herbivorous fish is significantly reduced on reefs without nearby seagrass and mangrove habitats (*Dorenbosch et al., 2005*; *Dorenbosch et al., 2007*).

Major barriers to the long-term survivorship of restored corals include disease and high levels of macroalgae that block coral recruitment (*National Marine Fisheries Service, 2015*). However, planting corals specifically near healthy mangrove and seagrass habitats could facilitate increased transplant growth (e.g., increased nutrient transfer) and survivorship (e.g., reduced disease and macroalgae due to herbivore biomass and diversity) (Fig. 3). Within reefs, corals could also benefit from being planted near larger, established corals due to wave protection and/or spillover of facilitated herbivores (i.e., urchins like *Diadema*) that live within the interstitial matrices of larger corals (Fig. 3). Despite the great potential for long distance positive interactions to enhance coral restoration, little consideration has been given to scientifically evaluate the effects of restoring corals with and without adjacent and healthy mangroves or seagrasses, or near large corals within reefs. These efforts could be further enhanced with more 'ridge to reef' management or protection (e.g., MPAs) that encompass all coastal habitats from land out to coral reefs.

# POSITIVE DENSITY DEPENDENCE

Positive density dependence is another mechanism that can enhance population recovery. The role of intraspecific density on a population's growth and success has been examined in a great number of organisms, including insects (e.g., *Stiling, 1987*), plants (e.g., *Harms et al., 2000*), and marine invertebrates (e.g., *Levitan, 1991*), and can have both negative and positive effects. Positive density dependence, or density-dependent facilitation, occurs when the success of conspecifics increases at higher densities (i.e., the Allee effect: *Allee, 1931*). For example, salt marsh grasses and mangrove trees have higher survivorship and increased rates of recovery in groups vs. in isolation because group benefits emerge from synergistic soil oxygenation and structural protection from wave stress (*Gedan & Silliman, 2009*; *Silliman et al., 2015*). Animal aggregations, such as mussels and schooling fish, have increased success with higher densities due to reduced predation pressure on individuals (*Gascoigne & Lipcius, 2004*). Recent ecological theory suggests that positive density dependent effects should be more prevalent under stressful physical conditions,

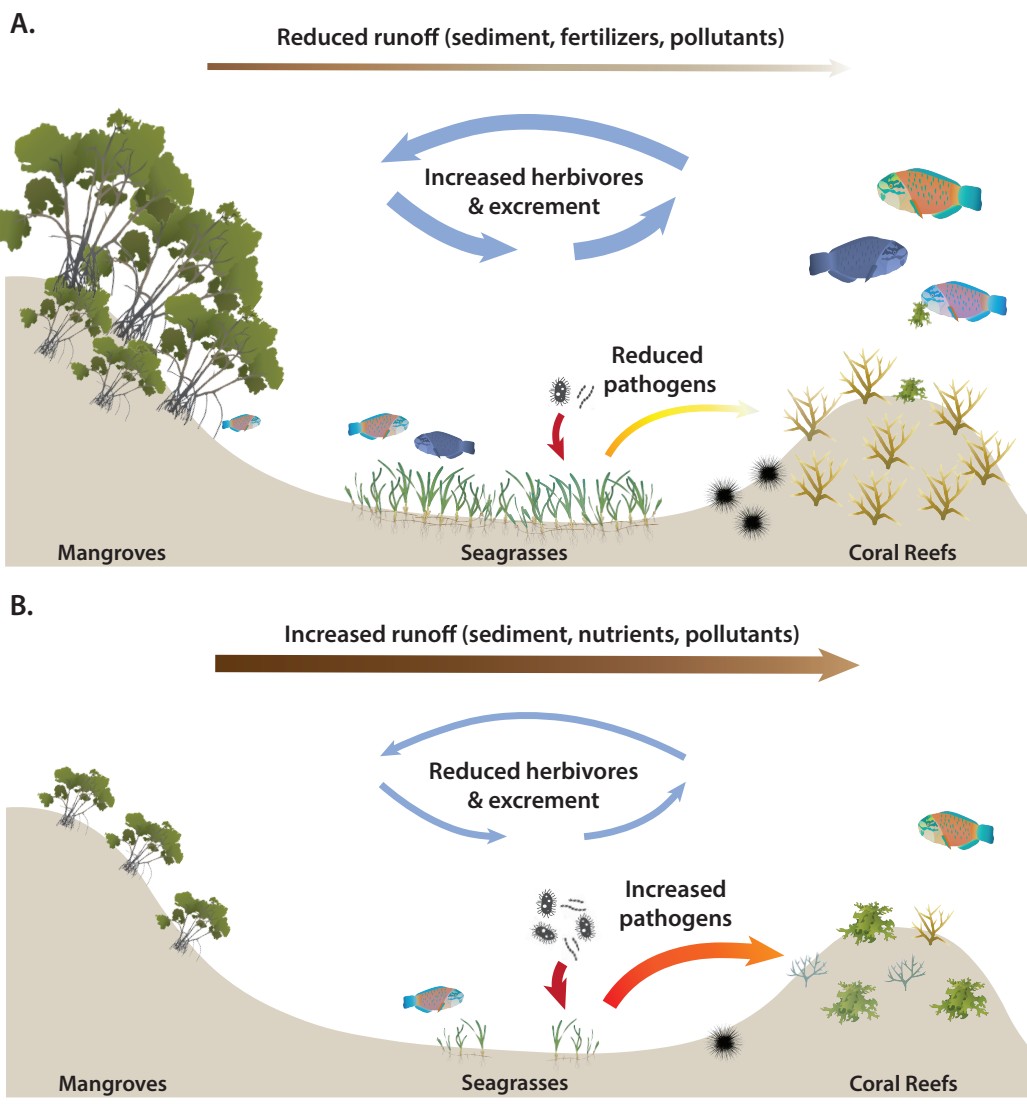

**Figure 3** **Potential effects of long-distance interactions on coral transplants (A) with and (B) without adjacent, healthy mangrove and seagrass habitats.** Brown lines indicate the movement of coastal or watershed-based run-off (which carry sediments, excess nutrients, and pollutants) from land that gets reduced by mangrove forests and seagrass beds before going on to coral reefs. Purple lines depict microbial cycling by seagrasses that reduces pathogens and coral disease. Thick lines indicate large effects or movements of particles, whereas thin lines indicate a small effects or movements of particles. (A) there is a abundance of herbivores, including greater diversity of parrotfish, due to nearby nursery habitat and food provisioning of mangroves and seagrasses, which leads to reduced macroalgae abundance on reefs. (B) there are fewer herbivores, more algae, disease, and run-off on reefs, leading to reduced success of coral transplants. Image credit (vector graphics): Catherine Collier, Jane Hawkey, Tracey Saxby, and Joanna Woerner, Integration and Application Network, University of Maryland Center for Environmental Science (http://ian.umces.edu/imagelibrary/).

while negative density effects like competition will occur more in low stress environments (i.e., the stress-gradient hypothesis, SGH: *Bertness & Callaway, 1994*). Because habitats being restored are often degraded areas with high physical stress, positive density effects may emerge and be important for improving restoration success.
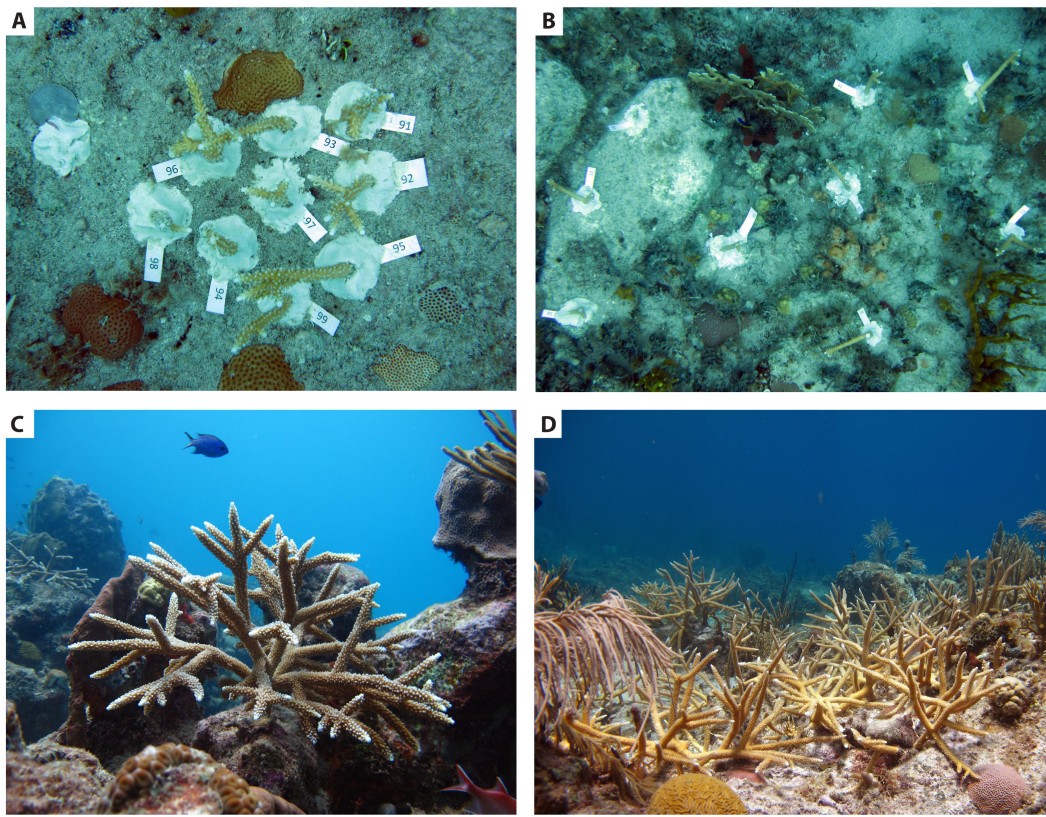

**Figure 4** **Manipulations of coral transplant density and spacing during restoration.** Experimental research by *Griffin et al. (2015)* in which Caribbean staghorn corals (*Acropora cervicornis*) were spaced close together (A) and far apart (B) (photograph was taken further away from corals in photo B). Photo credits: John Griffin, Swansea University, United Kingdom. Within a restoration program, restored corals are often planted in varying arrangements, for instance spaced apart in lower densities (C) or closer together to mimic a natural high-density acroporid thicket (D). Photo credits: Kemit Amon-Lewis, The Nature Conservancy. All pictures were taken in the US Virgin Islands.

Recent studies examining the effects of density on the success of conspecific coral transplants have found both negative and positive density dependence. The first study to manipulate density (corals spaced 5-cm apart) and planting configuration of staghorn coral (*Acropora cervicornis*) found increasing density significantly reduced coral growth within the first three months of transplantation, likely due to competition for space (*Griffin et al., 2015*) (Fig. 4). In the Philippines, *Shaish et al. (2010)* found no difference in colony survivorship of transplanted *Montipora digitata* in high density (spaced 10-cm apart) and low density (spaced 20-cm apart) plots over 15 months. These studies suggest that studies examining the effects of density on coral transplants may have varied results depending on spacing between colonies. For endangered Caribbean acroporid corals, recent experimental research shows a unimodal relationship may exist, with positive density effects occurring at moderate levels (3 corals m$^{-2}$) but negative density effects occurring at higher densities (*Ladd et al., 2016*). However, strong positive density effects have been found under natural settings in *A. cervicornis* thickets at high density levels due to a positive feedback between

these branching corals and the fish that live within their branches and provide nutrients through excrement (*Huntington et al., 2017*) (Fig. 4).

While most of these studies examine colony-specific metrics, such as growth and survivorship of individual corals, the effects of density are likely affected by environmental stressors and temporal scales. Although ecological theory (e.g., the SGH) suggests that positive density dependency may emerge under high stress conditions in coral reefs, no studies to date have examined how the impacts of density vary across a stress gradient. For instance, because of the tendency of branching corals to fuse, higher density plots may enhance structural resistance to wave energy such as intense storms (as suggested in *Griffin et al., 2015*), yet no study to date has examined this type of positive density effect. Similarly, higher density plots may also enhance structural complexity that attracts more fish (e.g., promoting nutrient transfer) or invertebrates (e.g., that promote corals through mutualisms or trophic facilitation) (Fig. 2). Clearly, further research on the role of density in promoting coral restoration is needed, specifically studies that focus on how density varies along realistic and commonly occurring stress gradients, like wave stress, heat stress, turbidity, and coral predation.

## POSITIVE LEGACY EFFECTS

Legacy effects, which can be both negative and positive, occur when species interactions have a lasting impact on an ecological community well past the time of the initial interaction. One example is the effect of ditching on salt marshes in the 1930's, where the immediate negative legacy effect of this human-ecosystem interaction included changes to plant distributions, fish and insect abundance, and hydrological patterns that benefited some species but suppressed others (*Silliman, Grosholz & Bertness, 2009*). Only recently, a positive legacy effect was found that ditched marshes are more susceptible to runaway grazing by crabs and subsequent community die-off because ditched marshes are dominated by plants that are preferred by grazing crabs (*Coverdale et al., 2013*). A legacy effect evident in the context of restoration includes oyster reefs, which have been decimated by disease and overharvesting over the last century (*Beck et al., 2011*). Although the individuals that built remnant oyster reefs decades ago are gone, their skeletons remain and are commonly used to facilitate oyster recruitment (i.e., 'spat') (*Schrack et al., 2012*).

Although legacy effects on coral reefs have not been intensely studied, we consider here one strong example that could be used in restoration. Like oyster reefs, coral skeletons remain present on reefs for many years after the veneer of living coral tissue has disappeared. In the same way that oyster spat recruits to old oyster shell, new coral recruits or transplants may also benefit from settling or growing on old skeleton. For instance, corals may be able to grow faster by re-sheeting tissue over the skeleton rather than expending energy on producing new skeleton. Indeed, a reef in Palau was found to recover quickly after a bleaching event because corals regrew tissue over former skeletons rather than laying down new skeleton (*Roff et al., 2014*). Growing on coral skeletons can also keep smaller, more susceptible coral transplants away from competitors or predators. For instance, coral larvae are known to avoid reefs with high algal cover, but will settle on structures

above the substrate to avoid competitive algae (*Dixson, Abrego & Hay, 2014*). Likewise, corallivores might prefer to feed at low elevations or avoid high elevations to potentially avoid predation (E Shaver, pers. comm., 2014–2015). The potential to use coral skeletons in a restoration context has recently been realized, for instance using new techniques (i.e., micro-fragging: *Forsman et al., 2015*) to restore reef-building boulder corals in Florida. Much more research is required, however, to understand and maximize the benefits of coral skeletons in coral restoration efforts and to identify other legacy effects that can be used to increase the efficiency of coral restoration efforts.

## BIODIVERSITY AND ECOSYSTEM FUNCTION

Biodiversity encompasses the species, genetic, functional, and ecological diversity of living things. Decades of research have shown that for a great variety of marine and terrestrial ecosystems, there is a positive relationship between biodiversity and ecosystem function, (*Tilman et al., 1997*; *Hooper & Vitousek, 1997*; *Kinzig, Pacala & Tilman, 2002*; *Hooper et al., 2005*; *Srivastava & Vellend, 2005*). For example, *Maestre et al. (2012)* found that diversity of plant species in semi-arid ecosystems (e.g., drylands) enhanced carbon and nutrient cycling as well as overall ecosystem multifunctionality. In addition to higher ecosystem productivity and functioning, studies have also shown that biodiversity increases ecosystem resilience to disturbance. Proposed mechanisms behind the positive impacts of biodiversity on ecosystem function and resilience include the asynchrony of species responses to environmental changes or disturbances (i.e., the portfolio effect: *Tilman, 1999*; *Loreau, 2010*) and increased functional redundancy, leading to higher resilience and the maintenance of ecological functioning if one or a few species are lost (i.e., the insurance hypothesis: *Yachi & Loreau, 1999*).

The most commonly used techniques for coral restoration today (e.g., coral gardening and outplanting: *Lirman & Schopmeyer, 2016*) were designed to restore branching corals in response to severe declines in the populations of two formerly dominant Caribbean acroporid corals after disease and bleaching events (staghorn coral: *A. cervicornis*; elkhorn coral: *A. palmata*; *Gardner et al., 2003*). Coral restoration projects across the globe also largely focus efforts on a few species of branching corals (e.g., *Acropora* spp. and *Pocillopora* spp.), due to a history of restoration methods that take advantage of their high growth rates and ease of propagation through fragmentation (*Johnson et al., 2011*; *Rinkevich, 2014*). However, coral reefs in both regions have a diversity of coral species and many massive coral species can be important reef-builders (e.g., *Orbicella* spp.). In addition, Caribbean branching corals can be more susceptible to natural enemies such as predators and disease (*Bruckner, 2002*; *Baums, Miller & Szmant, 2003*), and due to ecological trade-offs with competitive dominance, tend to be less tolerant of environmental stressors like warm temperatures and bleaching (*Loya et al., 2001*). Because natural enemies and climate change continue to impede restoration efforts (*National Marine Fisheries Service, 2015*) and because coral reefs naturally harbor a greater diversity of corals than just two species, reestablishing a range of coral biodiversity, including species and growth forms, may enhance success and facilitate ecosystem resilience.

Several experiments manipulating coral species richness have found improved coral growth and survivorship in mixed versus single species plots. For instance, *Dizon & Yap (2005)* found that colonies of *Porites cylindrica* had higher growth rates when transplanted in plots with other species (*Porities rus* and *Pavona frondifera*) relative to conspecifics, however this facilitation stopped when nutrient enrichment was introduced to plots. Diverse coral species facilitated *A. cervicornis* growth and survivorship by reducing the number of coral predators (*Coralliophila abbreviata*) that attacked plots of *A. cervicornis* (*Johnston & Miller, 2014*). Similarly, *Cabaitan, Yap & Gomez (2015)* found that transplants of the coral *Pavona frondifera* had higher success when planted with the coral *Porites cylindrica*, but only in stressful environments with high wave energy. In addition, *P. cylindrica* also appeared to reduced predation on *P. frondifera* by starfish and snails (*Cabaitan, Yap & Gomez, 2015*). Additionally, *Montoya-Maya et al. (2016)* found successful coral recruitment in areas transplanted with mixed coral species, though this study did not manipulate diversity and all species used were acroporid or pocilloporid (e.g., branching) corals. Importantly, restoring diverse functional traits (e.g., growth forms) in addition to species may also increase the resilience of restored sites to climate-change impacts (e.g., bleaching events or intense storms), as boulder corals may be more likely to withstand these stressors (*Loya et al., 2001*). Luckily, new techniques are being designed to improve the propagation of other coral species (*Lirman & Schopmeyer, 2016*). Future research should examine how coral diversity effects transplant success (growth, mortality, recruitment) as well as long-term ecological stability (resistance and recovery processes). These efforts may be especially important in the Caribbean, where it is thought that low diversity relative to the Indo-Pacific has reduced its ability to recover after disturbance.

## CONCLUSION AND RECOMMENDATIONS

Restoration is currently being elevated as a major conservation strategy to help combat coral loss in the Caribbean and across the globe. For this to be realized, the efficacy of restoration efforts must increase long-term coral transplant growth and survivorship while reducing associated costs. It is important to note that like other marine and terrestrial ecosystems, active transplantation efforts are likely to be successful only in sites where environmental or local stressors have been successfully reduced (e.g., effective MPAs) or in areas where stress is not an issue but recruitment limitation is impeding natural recovery (*Orth et al., 2012*). For new techniques and designs to help increase the scale of coral restoration efforts, these restorative efforts thus must also be tied to successful management strategies.

Harnessing naturally-occurring positive interactions in restoration designs has recently been suggested (*Halpern et al., 2007*; *Gedan & Silliman, 2009*) and demonstrated (*Silliman et al., 2015*) as one way scaling up restoration efforts can be accomplished in other aquatic habitats. Coral reef managers have used positive interactions for decades to promote corals with a well-known trophic facilitation by algae-eating herbivores, but have not yet identified other positive species interactions specifically for use in coral restoration designs. In this paper, we highlight several other general facilitative mechanisms that could be used by coral restoration practitioners when designing future projects. Although this paper is

focused on facilitations, the relative effects of negative (e.g., competition and predation) and positive interactions (facilitation and mutualism) are likely to vary depending on the species or organisms present and the environmental conditions or characteristics of that site (*He, Bertness & Altieri, 2013*). Nonetheless, below we recommend general facilitative mechanisms that deserve further testing and consideration in restoration designs, including:

(1) Identifying specific trophic linkages or food webs that promote coral health and transplanting corals in areas with where these food webs are robust and protected.
(2) Identifying mutualisms between corals and other reef- or coral-associated organisms that promote coral health and enhancing or protecting these mutualistic partners.
(3) Locating sites for coral restoration close to healthy and well-functioning seagrass and mangrove habitats and protecting these habitats from extractive or destructive uses.
(4) Identifying species-specific density effects for corals of restoration interest to take advantage of transplant densities that facilitate coral growth or survivorship.
(5) Exploring the legacy of former reef-builders or occupiers, such as coral skeletons left behind, for use in coral transplantation.
(6) Determining ways in which increased coral diversity facilitates the success of coral transplants as well as the long-term success of the restored coral reef community to environmental fluctuations and disturbances (e.g., resilience).

### Funding
This work was supported by Duke University. Duke University and the National Science Foundation supported EC Shaver and BR Silliman (EC Shaver: GRFP DGE 110640; BR Silliman: BIO-OCE 1056980). There was no additional external funding received for this study. The funders had no role in study design, data collection and analysis, decision to publish, or preparation of the manuscript.

### Grant Disclosures
The following grant information was disclosed by the authors:
Duke University and the National Science Foundation: GRFP DGE 110640, BIO-OCE 1056980.

### Competing Interests
The authors declare there are no competing interests.

### Author Contributions
- Elizabeth C. Shaver and Brian R. Silliman wrote the paper, prepared figures and/or tables, reviewed drafts of the paper.

### Data Availability
This article did not generate any data or code, as this manuscript reviews and synthesizes the current literature.

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
