# Peer review of "Time to cash in on positive interactions for coral restoration"

_PeerJ, doi:10.7717/peerj.3499_

## Round 0.1 · original submission · Minor Revisions

Both the reviewers and I believe this is an original and valuable contribution. If you could address most of the reviewer comments, and for those you don't agree with explain why that's the case, we should be good to go.

·

Basic reporting

No comment

Experimental design

1. The authors should include a brief overview of coral restoration project success rates in the introduction. This would strength the argument that incorporating positive interactions is needed.

2. The section on trophic facilitation does not address potential effects of predatory fishes (those that feed on herbivores). The goal of MPAs is to increase large predators as well as herbivorous fishes. This section would benefit from a discussion on how large predator recovery may influence coral restoration.

3. Positive density dependence: Lines 234-244. It is not clear why no density dependent effects were observed at the transplant colonies in the Philippines. What did the authors of that study conclude?

4. Positive legacy effects: Line 268. The authors should provide a transition from examples of negative legacy effects to positive legacy effects.

5. Biodiversity and ecosystem function: Line 301. Briefly explain insurance and portfolio hypotheses. This paper is relevant for a non-academic audience that might not be aware of or have access to the reference papers.

6. Biodiversity and ecosystem function: Line 336. Provide examples of the new techniques.

Validity of the findings

Provide support for the statement that restoration is a viable conservation strategy to mitigate coral loss at a regional scale. Based on the evidence presented, I am not convinced that coral restoration can mitigate the effects of climate change, nutrient loading, and overfishing. I agree that restoration can be a valuable tool and incorporating positive interactions is an important approach to improve the success of these projects. However, I think the authors need to spend a bit more time discussing the bigger picture. What are the limitations of restoration? What other conservation strategies are needed to facilitate restoration success?

Additional comments

The authors present a well written and valuable assessment of positive interactions that may facilitate coral restoration. I believe the authors can considerably improve the manuscript by providing more evidence that coral restoration (and incorporating positive interactions) can in fact mitigate coral decline when faced with the multitude of factors that continue to degrade reefs.

Reviewer 2 ·

Basic reporting

The manuscript “Time to cash in on positive interactions for coral restoration” by Shaver and Silliman is a nicely written review of a topic that has not been explored in coral reefs, at least no extensively if at all. The authors described half a dozen positive interactions that could be crucial in coral restoration. The manuscript is clearly written in professional and straightforward language with lack of grammatical errors.

Although the objective of the paper was to address positive interactions, there are also negative interactions that may affect coral restoration. I am thinking about coral-coral competition for space (e.g., the work of Allan Logan in the 1980s in Bermuda) and sponge-coral competition for space (Joe Pawlik work in the Bahamas and the Keys). This should be at least acknowledged because such interactions may override any positive association with other species.

Finally I am wondering if you should provide a list of concrete recommendations to coral restoration managers based on your review of the literature. For example, if you are asked by NOAA coral recovery planners to provide recommendations on how to do coral restoration or restocking for better, faster, and more economical coral recovery, what are those recommendations based on positive interactions? This will provide a more tangible objective to your paper and practicable outcome that could be used by directly by managers or at least mentioned in coral recovery plans. Otherwise, your paper will be another nice written review of the literature with a novel angle but you may have missed the opportunity to have a real conservation and applicable outcome.

Below are specific comments.

Abstract
L28 – I would say coral reefs have declined across the globe due to “human impacts and climate change”

L36- Change “enhance” for “promote” to avoid repeating the word twice in the sentence. For example: “that could promote population enhancement efforts” or just “that could enhance restoration”

Introduction
L55 – New paragraph starting in “Despite the….”

L59 – Please provide a specific web link for a coral restoration project within TNC, or a citation of a TNC report about coral reseeding, or from the Coral Reef Foundation restoration project, or even better a scientific paper evaluating reseeding success (e.g., Young et al 2012, Bull Mar Sci; Bright et al 2016, Restoration Ecology).

L65 – L77 – Please provide also an example of positive interaction in coral reef ecosystems that promote recovery. For example, crab associations with Pocillopora corals as you mention below (e.g., Stewart et al 2006, Coral Reef; Stier et al. 2010, Coral Reefs).

L85-86 – “Although coral restoration activities and programs have rapidly grown over the past decade (Lirman and Schopmeyer 2016)” This was already said in Line 60. Replace with different wording or eliminate.

L85-L91 – This is a ling sentence, I recommend splitting it in two ideas.

L93 – Please briefly explain what the 9.5% is based on. For example, “…from 1980-2016 shows “X” out “Y” studies (9.5%) have examined….” Same idea for the 60% part.

L114 – I would say “puffer fish” instead of puffers.

L116-117 – “sharks may facilitate corals through trophic interactions that enhance the abundance of urchins that reduce competitive macroalgae (Sandin et al. 2008).”I didn’t recall this suggestion from the Sandin et al 2008 PlosOne paper. I checked the paper again and they did not suggest this, at least directly. Please find a different study that supports this idea or eliminate this statement. For example, a clearer example (albeit from experimental study) involving sharks, triggerfish, urchins, and algae is the new paper by Jon Witman et al 2017 in PlosOne “Experimental demonstration of a trophic cascade in the Galápagos rocky subtidal: Effects of consumer identity and behavior”.

L125-127 “No restoration studies have used experimental or comparative methods to test the effects of transplanting corals in areas with high vs. low herbivore abundance on restoration success” But see this paper - > Miller et al 2014 MEPS Prevalence, consequences, and mitigation of fireworm predation on endangered staghorn coral

L132- Also add citation of Williams et al 2014 PeerJ Removal of corallivorous snails as a proactive tool for the conservation of acroporid corals

L205 – Perhaps a brief explanation of the status of Acropora in the Caribbean (especially in the FL keys) is necessary in this paragraph to put the topic into context. For example, explain that the main restoration work that is occurring in US waters and probably the rest of the Caribbean is the reseeding and fragment culture of the two species of Acropora because their populations declined drastically in the 1970 and 1980s due to WBD. Or just explain that are major barriers for coral restoration in general.

L212-214 – This idea is also exemplified by the current recommendations in Acropora Restoration. See Outplanting section, p-23 in Johnson et al 2011. Caribbean Acropora Restoration Guide: Best Practices for Propagation and Population Enhancement: 1-64 http://nsuworks.nova.edu/occ_facreports/71

L219 – I think a short lead sentence for this paragraph would add a more organized beginning. E.g., “Positive density dependence can enhance population recovery. The role of intraspecific….”

L236 – First time you mention Acropora cervicornis and should be spelled as full scientific name instead of A. cervicornis.

L247 – Also genotypic differences. See Drury et al. 2017 Genotype and local environment dynamically influence growth, disturbance response and survivorship in the threatened coral, Acropora cervicornis, PlosOne. https://doi.org/10.1371/journal.pone.0174000

L302 - effect’ <- the apostrophe is not necessary after effect

L303-306 – Lead sentence should be change since does not represent the narrative of the paragraph. The second sentence is a better choice. But even with this sentence, it is not clear to me the main topic of this paragraph. If you are talking that restoration in coral reefs needs diversity then a short version of the last sentence in the paragraph should be the lead sentence.

L307 – “restoration used today” <- the word used can be removed.

Conclusions

L354-355 – I think this review is missing a short and sweet list of concrete recommendations to coral restoration managers based on your review of the literature. E.g., how can you apply the info that supposed to be already the best available science on positive interactions for coral reefs to concrete coral reef restoration practices?

Experimental design

NA, This is review with not experimental design.

Validity of the findings

See basic reporting. This is a review paper. There are not original findings.

Additional comments

Good work.

---

## Round 0.2 · Minor Revisions

Thank you for your very rapid turnaround and for carefully considering the reviewer comments and outlining your responses to them. I read the revised manuscript and I think it is fabulous. It is VERY well written (super clear and concise). There is nothing like this out there for coral restoration. I was not familiar with the work on optimizing coral restoration, e.g., the work on Acropora density dependence. And I especially appreciated how you reviewed some of the relevant ecological theory and ecological and restoration work from other systems and tied that in with challenges in restoring coral populations. This is going to be a valuable and unique contribution (thanks for submitting it to PeerJ).

I have only one remaining suggestion and it is regarding how you frame the role of trophic cascades and tropical facilitation in regard to coral restoration. As I’m sure you know, there are strong differences of opinion among reef scientists about the functional roles of parrotfishes and sharks (and other top predators) on reefs. Particularly how they might indirectly affect corals and in this context coral transplants (meant for restoration). I suggest you highlight literature supporting both sides of the argument, pointing out there remains a lot of ambiguity, little data, and thus there is great need for more work. For example, in describing the role of sharks in facilitating coral population, you cite Sandin et al 2008. But the sample size of this study was only 4 reefs and the pattern was known before the team went out to document it (i.e., this was not an unbiased test of the relationship between corals cover and shark biomass). Studies with larger sample sizes and more randomized sampling have generally found no relationship or only a very weak positive relationship (e.g., Ruppert et al 2013 PLOS One, Williams et al 2015 PLOS One, Abel Valdivia’s unpublished surveys of Caribbean reefs - he also found no relationship). Moreover, it is widely believed that the dominant direction of effect is from the bottom up; i.e., corals have a positive effect on shark biomass via facilitation of great prey densities. So even when there is a positive association, it would be very hard to make a sound argument about the mechanism and direction of the interaction. Finally, many reef scientists are quite skeptical of trophic cascades via sharks, for a number of reasons, including their feeding on multiple trophic levels and their very broad foraging (see Valentine and Heck’s 2005 review of this: DOI 10.1007/s00338-004-0468-9).

Despite dozens of studies on how macroalgae affect the recruits of weedy coral taxa, I actually think there is a lot of ambiguity with the role of parrotfishes and even other grazers in facilitating or harming coral restoration. As many experimentalists know, parrotfishes love to bite transplanted coral nubbins! I had a massive experiment wiped out in the FL Keys when the top of nearly all my transplanted branches of Madracis were bitten off. Wellington and others have commented on this. Also please read the excellent review by Kuffner and Toth (http://onlinelibrary.wiley.com/doi/10.1111/cobi.12725/abstract also see Randi Rotjtn’s reviews on this subject doi: 10.3354/meps07531 ). They do a great job of contrasting the positive and negative effects of grazers on corals and reefs and the conclusion is if you consider parrotfish predation and erosion, it’s hard to argue for a net positive effect. My point is that it’s complicated and I think your review could be improved in this area by recognizing the complexity, and how much we don’t know, about species interactions in high diversity systems like reefs.

---

## Round 0.3 · accepted · Accept

Thanks so much for considering PeerJ for this review article and for being so fast and responsive in considering our suggested edits.